# Antibacterial Activity of Electrospun Polyacrylonitrile Copper Nanoparticle Nanofibers on Antibiotic Resistant Pathogens and Methicillin Resistant *Staphylococcus aureus* (MRSA)

**DOI:** 10.3390/nano12132139

**Published:** 2022-06-22

**Authors:** William B. Wang, Jude C. Clapper

**Affiliations:** Upper School, Taipei American School, 800 Chung Shan North Road, Section 6, Taipei 11152, Taiwan; jclapper@tas.tw

**Keywords:** copper, nanoparticles, polyacrylonitrile, electrospinning, Methicillin-Resistant *Staphylococcus aureus*, antibacterial activity

## Abstract

Bacteria induced diseases such as community-acquired pneumonia (CAP) are easily transmitted through respiratory droplets expelled from a person’s nose or mouth. It has become increasingly important for researchers to discover materials that can be implemented in in vitro surface contact settings which disrupt bacterial growth and transmission. Copper (Cu) is known to have antibacterial properties and have been used in medical applications. This study investigates the antibacterial properties of polyacrylonitrile (PAN) based nanofibers coated with different concentrations of copper nanoparticles (CuNPs). Different concentrations of copper sulfate (CuSO_4_) and polyacrylonitrile (PAN) were mixed with dimethylformamide (DMF) solution, an electrospinning solvent that also acts as a reducing agent for CuSO_4_, which forms CuNPs and Cu ions. The resulting colloidal solutions were electrospun into nanofibers, which were then characterized using various analysis techniques. Methicillin-Resistant isolates of *Staphylococcus aureus*, an infective strain that induces pneumonia, were incubated with cutouts of various nanocomposites using disk diffusion methods on Luria-Bertani (LB) agar to test for the polymers’ antibacterial properties. Herein, we disclose that PAN-CuNP nanofibers have successfully demonstrated antibacterial activity against bacteria that were otherwise resistant to highly effective antibiotics. Our findings reveal that PAN-CuNP nanofibers have the potential to be used on contact surfaces that are at risk of contracting bacterial infections, such as masks, in vivo implants, or surgical intubation.

## 1. Introduction

Airborne diseases are easily transmitted through respiratory droplets expelled from a person’s nose or mouth [1,2,3,4,5,6]. Recently, human beings have suffered from negative clinical outcomes due to the rapid transmission and high infectivity of these diseases [7,8]. Transmitters frequently excrete bacteria-containing microscale aerosol particles. Since these particles are extremely lightweight, they can easily disperse between carriers via air current diffusion [9,10]. Therefore, it has become increasingly important for researchers to discover materials that disturb or interrupt airborne bacterial transmission.

A prime example of an airborne bacterial disease is pneumonia, which is a common cause of health complications and death around the world [11,12,13]. Due to its high transmission rate, patients suffering from community-acquired pneumonia (CAP) develop acute lung infections caused by deposits of bacteria-containing aerosol particles in their alveoli [14,15,16,17,18]. The alveoli of infected individuals become filled with pus and fluid from tissue debris and dead blood cells [19,20,21]. It is common for patients to have trouble breathing due to limited oxygen intake and the inflammation or irritation of their lungs, eventually succumbing from asphyxiation [15,22,23,24].

One such common pathogen that causes CAP is *Staphylococcus aureus* (*S. aureus*) [25,26]. *S. aureus* is a gram-positive cocci bacteria that can travel through airborne pathways and into respiratory systems, causing patients to experience complications related to pneumonia [27,28]. Professionals in the medical and animal husbandry industries frequently use conventional antibiotics to counter bacteria like *S. aureus*, but over reliance on antibiotics usually results in bacterial mutation [29]. Recent increases in CAP morbidity have been attributed to the rapid spread of Methicillin-Resistant *S. aureus* (MRSA) bacteria strains, which have proven to be insusceptible to conventional antibiotic therapy [30,31,32]. The most common drug treatment for severe community-acquired MRSA (CA-MRSA) infections is vancomycin, but despite its success in neutralizing MRSA, Vancomycin Intermediate *S. aureus* (VISA) mutation strains have been discovered [33].

CAP-causing microbes tend to remain on high contact surfaces for extended periods of time if they are not eliminated [34,35,36,37]. It is common for community acquired diseases, especially lung infections like CAP caused by *S. aureus* variations, to reside atop high contact surfaces such as masks, handles, or clothing [38,39,40]. Hence, antimicrobial agents should be implemented onto these surfaces to exterminate harmful bacteria before they infect people. However, conventional antibiotics cannot be placed onto high contact surfaces because most antibiotics are used in vivo via direct injection or consumption through the use of pellets or tablets that contain antibiotics; in vitro surface antibiotics are relatively rare [41]. While conventional antibiotics are effective preventative measures set in place to kill disease-inducing pathogens once they enter the body, solid antimicrobial agents such as antimicrobial coatings or nanofibers should be developed to effectively counter CAP-causing microbes the moment they come into contact with patients or transmitters [42,43,44,45].

Carbon-based polymer nanofibers are commonly used in surface engineering applications due to their high surface area to volume ratio, malleability, and compact structure [46,47]. Among various organic carbon polymers, polyacrylonitrile (PAN) could be widely adopted for antimicrobial filtration due to effective fibril formation via electrospinning [48,49,50,51,52]. PAN has unique thermal stability properties that allows it to degrade before reaching its melting point [53,54,55]. Moreover, PAN is known for its strong mechanical characteristics and high carbon yield, making it the ideal substance for building a long lasting, solvent resistant antibacterial filter [56]. Metals such as gold, copper, and silver have been used in the past in fields such as medicine and environmental science as antimicrobial agents [57]. Copper (Cu) is an extremely accessible and relatively cheap material that is known to have antimicrobial properties and is also nontoxic to humans when consumed at low levels [58,59,60,61]. Moreover, Cu has a strong fixation stability on PAN; other studies have also indicated that Cu leaching in water from carbon nanofibers is often negligible [48]. By synthesizing copper nanoparticles (CuNPs) onto carbon-based nanofibers, CuNPs and its subsequent Cu ions formed during the reduction of CuSO_4_ via *N,N*-Dimethylformamide (DMF) can easily permeate into bacteria [62,63,64]. Previous studies indicate that DMF, a common reagent and solvent used in colloidal synthesis, reduces Cu^2+^ to atomic CuNPs, which, as seen in X-ray Photoelectron Spectroscopy (XPS) tests shown in Appendix A, could then be later oxidized if it comes into contact with air [65,66]. Due to its high oxidation potential, DMF has the capability of reducing metal salts and forming nanoparticles during colloidal solution formation [67]. When synthesized into nanomaterials via reduction and electrospinning, CuNPs undergo significant physiochemical changes that allows them to have better permeability in pathogens than their bulk counterparts due to their size-dependent crystalline structure and high surface area-to-volume ratio [68]. The electrostatic interactions between the positively charged copper ions and the negatively charged peptidoglycan-based bacteria cell wall ruptures the cell wall via depolarization and releases internal cell contents [62,69]. Biochemical processes inside the cell are also disrupted when Cu ions originating from CuNPs interact with sulfur (S) containing biomolecules, replacing their respective H^+^ groups and in turn disrupting their molecular structure [62,70]. Moreover, CuNPs and their corresponding ions create reactive oxygen species (ROSs) when interacting with bacteria, which depletes intracellular ATP production and disrupts DNA replication [62,71]. All of these antibacterial processes result in the subsequent death and degradation of the targeted pathogen (Figure 1).

Electrospinning techniques are often used to synthesize carbon-based polymer nanofibers coated with nanoparticles [72]. DMF can be used as an electrospinning solvent for PAN and a reducing agent for CuSO_4_, which induces the chemical formation of CuNPs [73]. The electrospinner ejects PAN/DMF/CuNP solution from a nozzle and uses a high voltage electric field gradient to spin, solidify, and coagulate the solution into a solid PAN-CuNP nanofiber filament (Figure 2) [74].

## 2. Materials and Methods

### 2.1. Preparation of PAN/DMF/CuNP Solution

Prior to synthesizing CuNP-coated PAN nanofibers with an electrospinner, different colloidal electrospinning solutions were prepared. Four distinct PAN solutions were prepared by dissolving 10 wt.% of PAN (Merck Co., Ltd., Sigma-Aldrich Company, Neihu, Taipei, Taiwan) in 50 mL DMF (Merck Co., Ltd.). Different percentages (5%, 10%, and 15% wt.% w.r.t to weight of PAN) of CuSO${}_{4}$ (Merck Co., Ltd.) were simultaneously dissolved in three of the solutions, with the fourth solution being left as a pure PAN-based control that will not contain CuNPs. The solutions were then stirred using a 50 mm × 8 mm magnetic stir bar (Merck Co., Ltd.) at 200 rpm for 24 h. After the chemicals have homogenized and completely dissolved, a color change from light blue to dark green can be observed (Appendix A), qualitatively screening for and signifying the formation, oxidation, and agglomeration of CuNPs in solution [75].

### 2.2. Electrospinning

5 mL of PAN/DMF/CuNP solution created a specific concentration of CuSO_4_ (0%, 5%, 10%, and 15% wt.% w.r.t to weight of PAN) was loaded into a 10 mL single use Luer Slip Syringe (Terumo Co., Ltd., Shibuya City, Tokyo, Japan). The syringe was then affixed to a syringe pump (Inovenso Co., Ltd., Istanbul, Turkey) and connected to an Inovenso Basic System electrospinner (Inovenso Co., Ltd.) via a single use plastic tube that is attached to an electrospinning nozzle (Inovenso Co., Ltd.). A 200 × 200 mm${}^{2}$ piece of aluminum foil was attached onto to the movable collection platform of the electrospinner, which was locked in a position 100 mm away from the electrospinning nozzle. The negative electrode clip was then attached to the aluminum foil to allow for the creation of an electric field during the electrospinning process.

The electrospinner was set to operate at a voltage of 30 kV, and the injection rate was adjusted to 2.5 mL/h. Electrospinning concluded once the precursor solution was completely used up and spun into PAN nanofibers.

### 2.3. Bacterial Culture Preparation and Serial Dilution

Bacteria media and growth plates were prepared prior to growing various bacteria strains. 2.5% wt.% Luria-Bertani (LB) powder (Merck Co., Ltd.) was dissolved in distilled water and sterilized using an autoclave to create the growth media for all bacteria cultures used in this study. Similarly, 2.5% wt.% Luria-Bertani (LB) powder and 1.5% wt.% agar powder (Merck Co., Ltd.) was dissolved in distilled water, sterilized in an autoclave, and evenly poured into 100 × 15 mm polystyrene petri dishes (Alpha Plus Scientific Co., Ltd., Longtan District, Taoyuan City, Taiwan). LB-agar nutrient plates were formed after the solution solidified in the petri dishes.

Cultures of MRSA, MRSA Staphylococcal Cassette Chromosome *mec* (SCC*mec*) type II, MRSA SCC*mec* type III, MRSA SCC*mec* type IV, MRSA SCC*mec* type V_T_, VISA, *S. aureus*, *S. epidermidis*, *S. agalactiae*, *S. pneumoniae*, *E. faecalis*, *K. pneumoniae*, and *E. coli* (Bioresource Collection and Research Center, Hsinchu, Taiwan) were then grown in 2.5% LB broth in a shaking incubator (Thermo Fisher Scientific, Waltham, MA, USA) set at 37 ${}^{\circ }$C and 200 rpm for 24 h. The cultures were then diluted to a 0.5 MacFarland bacterial turbidity standard with a UV-VIS optical density spectrophotometer (Vernier Software & Technology, Beaverton, OR, USA), which provides an optical density comparable to the density of a bacterial suspension with a 1.5 × 10^8^ colony forming units (CFU/mL). 50 μL of different bacteria was added to each plate and glass beads (Merck Co., Ltd.) were used to equally distribute the bacteria on the plate.

### 2.4. Zone of Inhibition Antibacterial Tests

A 6.5 mm diameter hole puncher (Long Jer Precise Industry Co., Ltd., Taichung, Taiwan) was used to cut out all nanofiber and control disks. Three fiber disks of diameter 6.5 mm for each of the four concentrations (0%, 5%, 10%, and 15% wt.% w.r.t to weight of PAN) of PAN-CuNP nanofiber was placed onto a LB-agar plate for every bacteria strain. Three pure bulk copper disks of the same diameter were also cut out from a piece of pure copper foil and were used in Zone of Inhibition (ZOI) antibacterial tests under the same conditions for all bacteria strains.

The plates were then incubated with the fiber disks at 37 ${}^{\circ }$C for 24 h in a non-shaking incubator (Deng Yng Co., Ltd., Taishan District, Taipei, Taiwan). After incubation, the inhibition diameter of the various fiber and control disks were measured with ImageJ (Wayne Rasband, National Institutes of Health, Bethesda, MD, USA), a Java-based image processing program commonly used to analyze ZOI tests.

### 2.5. PAN-CuNP Nanofiber Characterization Techniques

3D topographical images of the fibers were obtained by using a XE7 Atomic Force Microscope (AFM) (Park Systems, Suwon-si, Korea) to identify the surface morphology of the fibers and the CuNPs coated on them. The AFM images were analyzed using the XEI imaging software (Park Systems), a Java-based image processing program exclusively designed for XE Atomic Force Microscopy.

A Phenom ProX G6 Desktop Scanning Electron Microscope (SEM) (Thermo Fisher Scientific) was used in the study to determine fiber morphology and thickness. Energy Dispersive X-ray (EDX) elemental analysis was also conducted in the SEM to analyze the elemental contents of the nanofibers.

The size of CuNPs that were suspended in PAN-DMF colloidal solution were measured with length characterization tools with a Talos F200X G2 Transmission Electron Microscope (TEM) (Thermo Fisher Scientific). 15% PAN/DMF/CuNP colloidal solution was coated onto a copper grid at a thickness of 100 nm and analyzed with TEM techniques. EDX elemental analysis was conducted in the TEM to analyze the elemental contents of the nanoparticles in the colloidal solution. In addition, dynamic light scattering (DLS) (Beckman Coulter, Inc., Brea, CA, USA) was also used to determine the size distribution of CuNPs suspended in PAN-DMF solution. 1 mL of 15% PAN/DMF/CuNP colloidal solution was loaded into different 12 mm square polystyrene cuvettes (Alpha Plus Scientific Co., Ltd.) and the samples were analyzed with a 4 mW He–Ne laser (Beckman Coulter, Inc.) operating at 633 nm with a scattering angle of 173${}^{\circ }$ on a N5 Submicrometer Particle Size Analyzer.

## 3. Results

### 3.1. Morphology Analysis of PAN-CuNP Nanofibers with AFM

The AFM is a powerful non-optical imaging technique used for surface analysis [76]. Pure PAN nanofiber disks and PAN-CuNPs nanofiber disks were analyzed using an AFM to determine their topographical structure. Both images were obtained in a 2500 μm^2^ frame with a non-etching AFM cantilever at a Scan Rate of 0.5 Hz (Figure 3).

PAN nanofibers have a compact structure, as they were found to scaffold atop each other and frequently intersect. As shown in Figure 3a, nanoparticles were not visible on pure PAN nanofibers. On the other hand, the image verifies that the PAN-CuNP nanofibers were coated with nanoparticles, as individual nanoparticles can be observed as small bumps on the surface of a PAN nanofiber (Figure 3b).

### 3.2. Elemental Analysis of PAN-CuNP Nanofibers with SEM EDX Spectra

The presence of CuNPs on PAN nanofibers was further investigated using SEM techniques, as shown in Figure 4. As depicted in the figure via SEM elemental composition scattering, it can be observed that CuNPs are scattered atop the surface of PAN nanofibers. Results from EDX analysis, as shown in Figure 5 and Table 1, reinforce that increased molecular weight percentage of Cu can be found in PAN-CuNP nanofibers synthesized from a higher wt.% of CuSO_4_.

It has also been observed that PAN-CuNP nanofibers synthesized from increasingly higher wt.% of CuSO_4_ have decreased nanofiber diameter (Table 1). This is a direct result of increased charge density from the heightened concentration of CuNPs in the based PAN-DMF solution. The increased charge gradient in the colloidal solution induces stronger elongation forces when acted upon with an electrospinner, thus resulting in decreased nanofiber diameter [77].

### 3.3. Characterization of CuNP Size with TEM and DLS

CuNPs found on PAN were also characterized for their size using HAADF-TEM analysis. An example of this is shown in Figure 6a, which depicts the TEM image of multiple CuNPs. Isolated, non-clustered CuNPs were observed to be scattered in 15% PAN/DMF/CuNP colloidal solution and measured for their size. The CuNPs analyzed using TEM techniques had relatively spherical structures that varied in size and shape. The elemental distribution of copper (Figure 6b) and its subsequent EDX Spectrum (Figure 7) also demonstrates that the particles imaged with the TEM were indeed CuNPs as high intensities of copper were measured in regions that contained the particles. The other elemental distributions of the same TEM image can be found in Appendix A.

Dynamic Light Scattering was also used to examine size distributions of CuNPs found in 15% PAN/DMF/CuNP colloidal solution. Results from DLS further reinforced the CuNPs diameter measurements obtained using TEM techniques (Figure 8). Larger particle measurements are noted as a possible experimental uncertainty and could have been incidences of clusters of CuNPs being identified together as a singular particle. As all PAN/DMF/CuNP colloidal solutions were mixed at a constant speed for the same amount of time, the variation in CuNPs sizes was presumed to be controlled and similar amongst solutions synthesized from different concentrations of CuSO_4_ [78,79].

### 3.4. Antibacterial Efficiency Tests

*Escherichia coli* (*K-12 DH5α*) was initially used for PAN-CuNP nanofiber antibacterial efficiency tests due to the bacteria’s ubiquitous nature and optimal growth kinetics. As seen in Table 2 and Appendix A, pure PAN nanofiber and bulk Cu disks do not have any antibacterial properties. On the other hand, PAN-CuNP nanofibers synthesized from a higher wt.% of CuSO_4_ showed significant antibacterial efficiency, with gradually increasing ZOI Diameter Measurements.

PAN-CuNP nanofiber disks were then tested on common disease inducing BSL-2 bacteria with antibiotic resistance for their antibacterial efficiency in comparison to *E. coli*. These six bacteria strains were selected because of their high infectivity and their ubiquitous nature. The key characteristics of these bacteria species and their related diseases are listed below in Table 3.

As shown in Table 4 and Appendix A, 15% PAN-CuNP nanofiber disks were generally more effective than their 10% PAN-CuNP nanofiber disk counterparts. Furthermore, it is also important to note that four of these strains (*S. epidermidis*, *S. agalactiae*, *E. faecalis*, *K. pneumoniae*) were completely resistant to ampicillin, a conventional antibiotic used to treat bacterial infections. Moreover, both 10% PAN-CuNPs nanofiber and 15% PAN-CuNP nanofibers were more effective than ampicillin disks for every bacteria other than *S. pneumoniae*. PAN-CuNP nanofibers also demonstrated the most antibacterial activity against *S. aureus* (*10780*) out of these six types of bacteria.

PAN-CuNP nanofiber disks were then placed on six differently genotyped MRSA and VISA strains (MRSA SCC*mec* type II, MRSA SCC*mec* type III, MRSA SCC*mec* type IV, MRSA SCC*mec* type V_T_, VISA) and tested for their antibacterial activity. *S. aureus* infections are usually treated with penicillin; in the event where penicillin fails to treat the bacterial infection, methicillin and vancomycin will be used to eliminate the bacteria [80,81,92]. However, mutated strains are difficult to eliminate with conventional antibiotics and are more capable of inducing CAP related infections compared to non-antibiotic resistant *S. aureus* [93]. This is because mutated *S. aureus* strains such as MRSA are known to have a multitude of genotyping characteristics that require varying methods of treatment [94,95,96].

Therefore, as seen in Table 5, Figure 9 and Appendix A, 15% PAN-CuNP nanofibers were tested for their efficiency on the five aforementioned variations of Methicillin-Resistant *S. aureus* and one strain of Vancomycin-Intermediate *S. aureus*. The data indicates that 15% PAN-CuNP nanofiber disks demonstrate antibacterial activity against MRSA and VISA.

## 4. Discussion

AFM, SEM, TEM, and DLS tests successfully confirmed the formation of CuNPs on scaffolded PAN-CuNP nanofibers.

PAN-CuNP nanofibers synthesized from PAN/DMF/CuNP colloidal solution via electrospinning have shown antibacterial effiency against various strains of bacteria. Nanofibers created from higher wt.% concentrations CuSO_4_ showed higher antibacterial effiency against bacteria strains. Moreover, PAN-CuNP nanofiber were more effective than pure PAN nanofibers and bulk Cu disks, demonstrating the antibacterial properties of CuNPs. Furthermore, PAN-CuNP nanofiber demonstrated antibacterial activity against Methicillin-Resistant and Vancomycin-Intermadiate *S. aureus* variants, bacteria strains that were otherwise immune to powerful conventional antibiotics. This implies that these nanofibers could become an alternative to these antibiotics when it comes to in vitro applications. 15% PAN-CuNP nanofibers demonstrated antibacterial activity against all 13 strains of bacteria used in this study despite the fact that each pathogen possesses different characteristics.

Conventional antibiotics are usually more expensive and stored under more specific conditions compared to carbon nanofibers coated with CuNPs, meaning that PAN-CuNP nanofibers can be used as a new antibacterial material that is also more cost-effective and easier to store in certain industries or fields of study [97,98]. In the future, PAN-CuNP nanofibers could be engineered or implemented onto high contact surfaces such as masks and medical or surgical equipment like, implants, tubes and catheters [99]. As copper is a trace element in the human body, PAN-CuNP nanofibers should be a safer option for in vivo or in vitro treatment as compared to other metals that are more toxic or harmful [100]. Hence, it would be possible for these nanofibers to eliminate or reduce bacteria upon immediate contact before they enter other organisms, cause irreversible infections, or mutate into antibiotic resistant.

## Figures and Tables

**Figure 1 nanomaterials-12-02139-f001:**
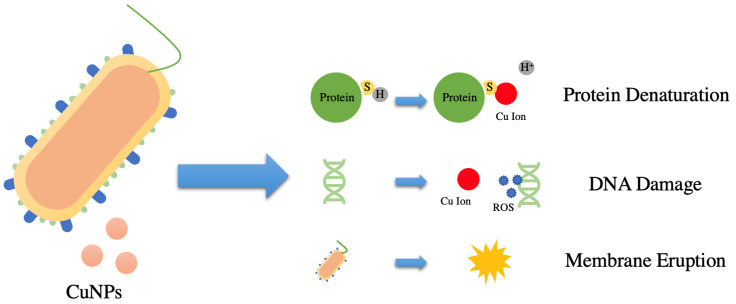
Schematic Diagram Demonstrating the Antibacterial Mechanism of CuNPs and Cu Ions, which Induces Protein, DNA, and Cell Membrane Damage in Bacteria Cells.

**Figure 2 nanomaterials-12-02139-f002:**
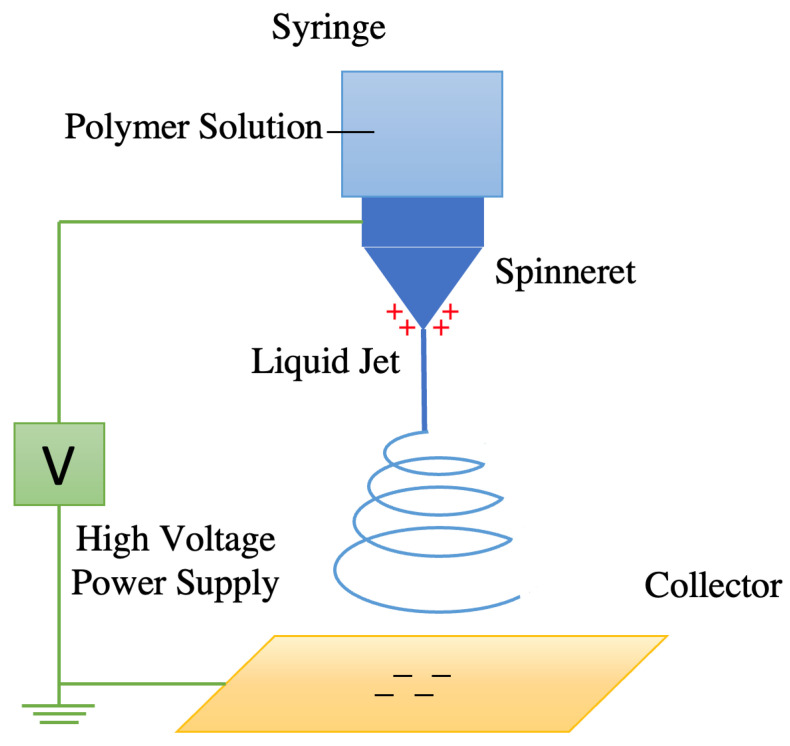
Schematic Diagram of Electrospinner Creating Carbon-Based Polymer Nanofibers.

**Figure 3 nanomaterials-12-02139-f003:**
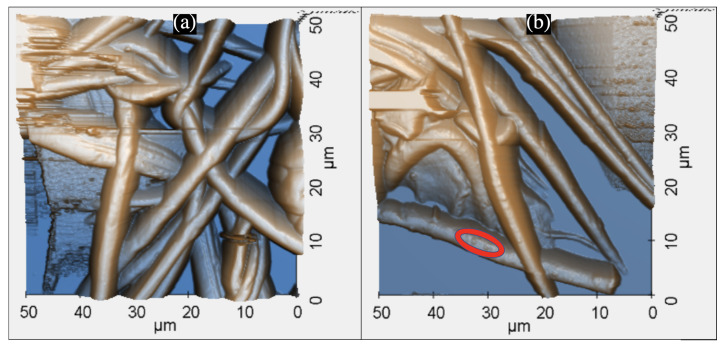
(**a**) 3D AFM Scanning Image of Pure PAN Nanofiber Disk (**b**) 3D AFM Scanning Image of 15% PAN-CuNP Nanofiber Disk, where Circled Portions Show Evidence of Nanoparticle Formation.

**Figure 4 nanomaterials-12-02139-f004:**
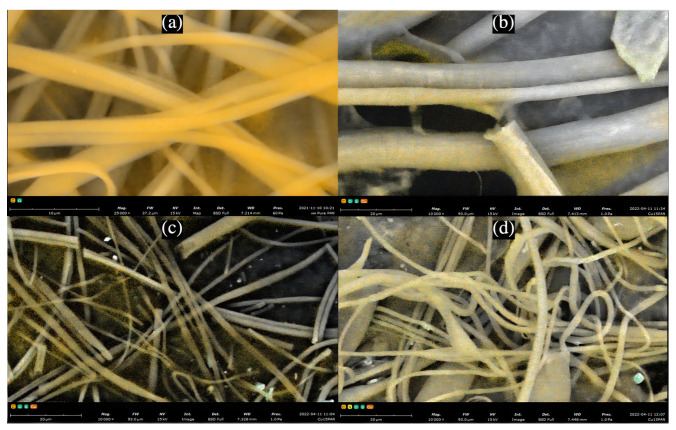
(**a**) SEM Elemental Distribution of Pure PAN Nanofibers (**b**) SEM Elemental Distribution of 5% PAN-CuNP Nanofibers (**c**) SEM Elemental Distribution of 10% PAN-CuNP Nanofibers (**d**) SEM Elemental Distribution of 15% PAN-CuNP Nanofibers.

**Figure 5 nanomaterials-12-02139-f005:**
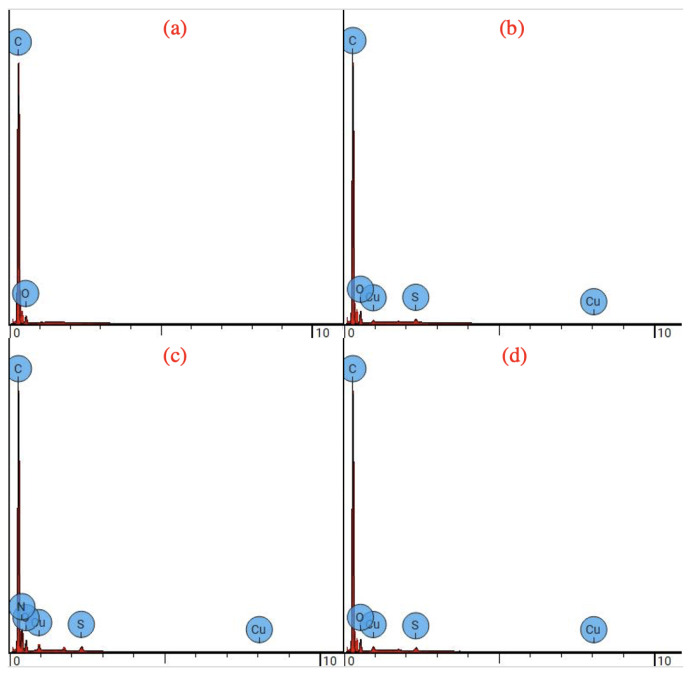
(**a**) EDX Spectrum of Pure PAN Nanofibers (**b**) EDX Spectrum of 5% PAN-CuNP Nanofibers (**c**) EDX Spectrum of 10% PAN-CuNP Nanofibers (**d**) EDX Spectrum of 15% PAN-CuNP Nanofibers.

**Figure 6 nanomaterials-12-02139-f006:**
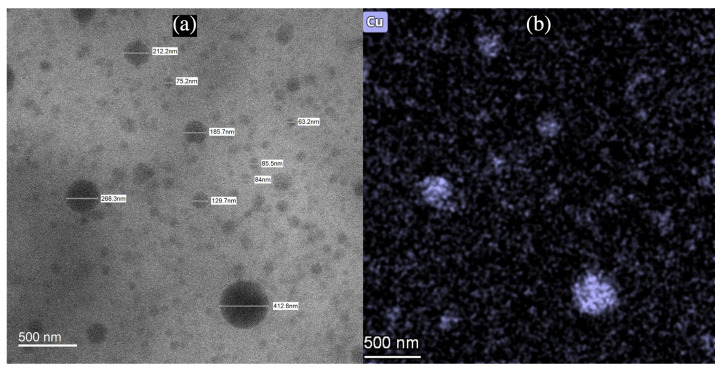
(**a**) TEM Image of CuNPs of Various Sizes in 15% PAN/DMF/CuNP Colloidal Solution (**b**) Elemental Distribution of Copper in TEM Image of 15% PAN/DMF/CuNP Colloidal Solution.

**Figure 7 nanomaterials-12-02139-f007:**
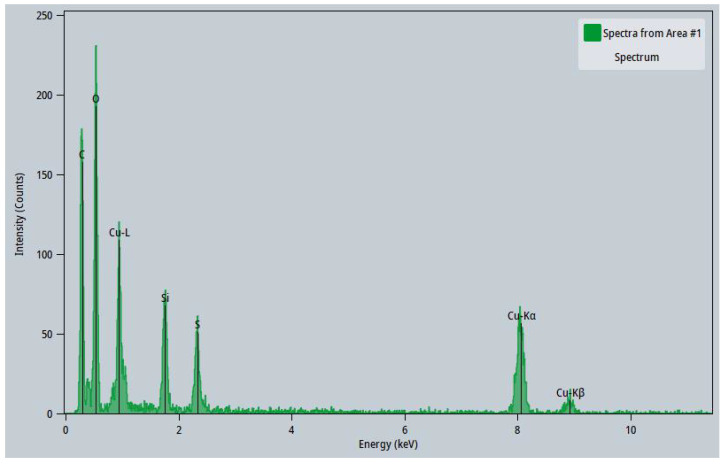
EDX Spectrum of 15% PAN/DMF/CuNP Colloidal Solution.

**Figure 8 nanomaterials-12-02139-f008:**
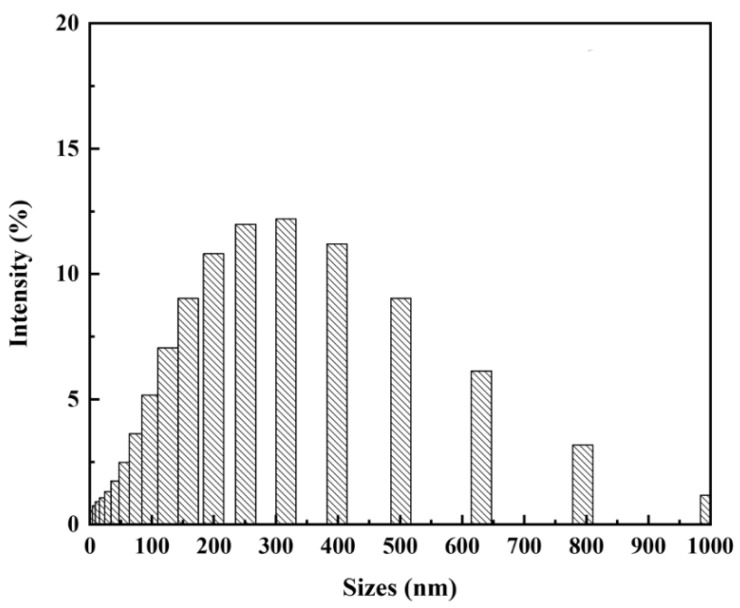
Dynamic Light Scattering Spectra of 15% PAN/DMF/CuNP Colloidal Solution.

**Figure 9 nanomaterials-12-02139-f009:**
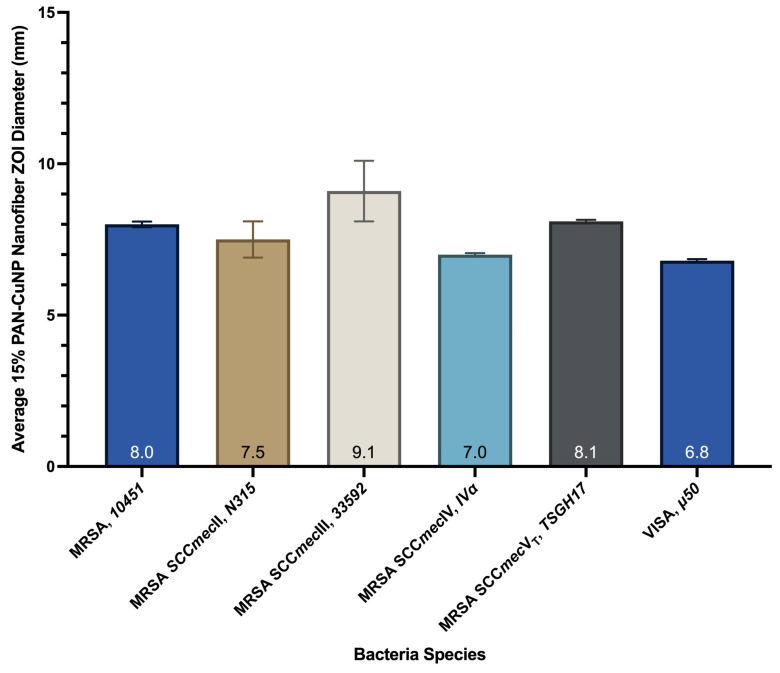
Bar Graph Representation of the Antibacterial Efficiency of 15% PAN-CuNP Nanofibers on Various MRSA and VISA Strains.

**Table 1 nanomaterials-12-02139-t001:** Average Nanofiber Diameter and Average Copper Weight Concentration for PAN-CuNP nanofibers Synthesized from Different wt.% of CuSO_4_ Over Five Trials.

Sample Classification	Average Nanofiber Diameter (nm)	Average Cu Weight Concentration (%)
Pure PAN Nanofiber	5124	0
5% PAN-CuNP Nanofiber	3617	1.38
10% PAN-CuNP Nanofiber	1398	1.72
15% PAN-CuNP Nanofiber	552	2.29

**Table 2 nanomaterials-12-02139-t002:** Antibacterial Efficiency of PAN-CuNP nanofibers Synthesized from Different wt.% of CuSO_4_ and Bulk Copper Disks on *E. coli*.

Bacteria Species	Sample Classification	Average ZOI Diameter (mm)	StDev (Over 3 Trials)
*E. coli* (*K-12 DH5α*)	Pure PAN nanofiber	0	0
*E. coli* (*K-12 DH5α*)	Bulk Cu Disk	0	0
*E. coli* (*K-12 DH5α*)	5% PAN-CuNP nanofiber	7.5	0.01
*E. coli* (*K-12 DH5α*)	10% PAN-CuNP nanofiber	8.0	0.01
*E. coli* (*K-12 DH5α*)	15% PAN-CuNP nanofiber	8.6	0.02

**Table 3 nanomaterials-12-02139-t003:** Key Characteristics and Related Diseases of BSL-2 Bacteria Tested in the Present Study.

Bacterial Species	Key Characteristics	Related Diseases
*S. aureus* (*10780*) [80,81]	Opportunistic pathogen, ubiquitous commensal bacterium, some strains have methicillin resistance (MRSA) or vancomycin resistance (VRSA)	Pneumonia, Cellulitis, Bacteremia, Endocarditis
*S. epidermidis* [82,83]	Opportunistic pathogen, occasional appearance at implant sites, highly resistant to antibiotics	Nosocomial sepsis, Endocarditis, Osteomyelitis, Peritonitis
*E. faecalis* (*10066*) [84,85]	Normal flora of gastrointestinal tracts, some strains have vancomycin resistance (VRE)	Urinary tract Infection, endocarditis, Inflammatory Bowel Diseases, Periodontitis
*S. agalactiae* (*10787*) [86,87]	Colonizes the genital tract of some women, causing vertical transmission	Neonatal sepsis, meningitis, pneumonia
*S. pneumoniae* [88,89]	Respiratory pathogen, some strains have antibiotic resistance	Pneumonia, Bacteremia, Meningitis, Otitis Media, Sinusitis
*K. pneumoniae* [90,91]	Gram-negative bacterium, respiratory pathogen, urinary tract pathogen, some strains have antibiotic resistance	Pneumonia, Urinary Tract infection, Nosocomial Bacteremia

**Table 4 nanomaterials-12-02139-t004:** Antibacterial Efficiency of PAN-CuNP nanofibers Synthesized from Different wt.% of CuSO_4_ on Various Bacteria Strains.

Bacteria Species	Average Ampicillin ZOI Diamter (mm)	Average 10% PAN-CuNP Nanofiber ZOI Diamter (mm)	Average 15% PAN-CuNP Nanofiber ZOI Diamter (mm)
*S. aureus* (*10780*)	6.8	9.3	9.7
*S. epidermidis*	0	6.7	8.3
*E. faecalis* (*10066*)	0	6.7	7.1
*S. agalactiae* (*10787*)	0	9.3	9.4
*S. pneumoniae*	8.3	6.7	7.2
*K. pneumoniae*	0	7.3	8.1

**Table 5 nanomaterials-12-02139-t005:** Antibacterial Efficiency of 15% PAN-CuNP Nanofibers on Various MRSA and VISA Strains.

Bacteria Species	Sample Classification	Average 15% PAN-CuNP Nanofiber ZOI Diamter (mm)	STDev (Over 3 Trials)
Methicillin-Resistant *S. aureus* (MRSA, *10451*)	15% PAN-CuNP nanofiber	8.0	0.09
Methicillin-Resistant *S. aureus* (MRSA SCC*mec*II, *N315*)	15% PAN-CuNP nanofiber	7.5	0.60
Methicillin-Resistant *S. aureus* (MRSA SCC*mec*III, *33592*)	15% PAN-CuNP nanofiber	9.1	1.00
Methicillin-Resistant *S. aureus* (MRSA SCC*mec*IV, *IVa*)	15% PAN-CuNP nanofiber	7.0	0.05
Methicillin-Resistant *S. aureus* (MRSA SCC*mec*V_T_, *TSGH17*)	15% PAN-CuNP nanofiber	8.1	0.05
Vancomycin-Intermediate *S. aureus* (VISA, μ*50*)	15% PAN-CuNP nanofiber	6.8	0.05

## Data Availability

The data supporting this study’s findings are available from the corresponding author upon reasonable request.

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
