# Peer review of "Antibacterial Activity of Electrospun Polyacrylonitrile Copper Nanoparticle Nanofibers on Antibiotic Resistant Pathogens and Methicillin Resistant Staphylococcus aureus (MRSA)"

_nanomaterials, 2022, doi:10.3390/nano12132139_

Round 1

Reviewer 1 Report

This work shows antibacterial activity of electrospun polyacrylonitrile mixed with copper nanoparticles. The use of PAN and Cu NP composites for antibacterial surface generation has been well established: e.g., Polymer Bulletin volume 77, pages4197–4212 (2020); Colloid Polym Sci (2015) 293:2525–2530; Polymers, 2022, 14, 1661 (and many others in the literature). Hence the work is not novel. The readability of figures is severely poor.

Reviewer 2 Report

This manuscript describes the preparation of polyacrylonitrile-based nanofibers coated with different concentrations of copper nanoparticles (PAN-CuNPs), their characterization by different techniques (AFM, TEM, SEM, EDX, DLS) and their antibacterial activity. The idea of developing antibiotic surfaces based on CuNPs is interesting, also taking into account that many bacteria have become resistant to some of the common used antibiotics. I think that the manuscript fits the scope of the journal and might be published after revision as follows:

1) Although the PAN-CuNPs have been characterized by different means, there are some issues that remain unclear to me:

- Is the PAN-CuNPs nanofiber just a solid heterogeneous mixture of PAN and CuSO4?

- The oxidation state of Cu has not been defined. Is it Cu(II)?; please, see my comment on page 4 below. I suggest to perform XPS analyses to determine the oxidation(s) state(s) of Cu, maybe also of S.

- In order to potentially being applied as an antibiotic surface, the fixation of Cu on the nanofiber should be tested. What happens, for instance, when the nanofiber is immersed into water? Is part of the Cu leached into the solution? Cu leaching could be measured by ICP-MS or AAS.

2) Page 1:

- The text corresponding to refs. 1-5 is of a general topic and, therefore, the references should be reviews and/or monographs whenever possible.

- Ref. 17 and 22 look too old (1938); maybe they can be changed for a more recent ones, unless they are milestones in the field.

3) Page 2:

- Refs. 55-57. The non-toxicity of CuNPs in vitro has also been reported (see, for instance: Nanotoxicology 2020, 14 (5), 683-695.

- Ref. 61 does not seem related with the corresponding text in the introduction.

4) Page 3:

- Figure 2: some text is too small to be readable.

- “N-N Dimethylformamide” must be “N,N-Dimethylformamide” with “N” in italics.

5) Page 4: The authors describe a color change from colorless to dark green as a sign of CuNPs formation, something that it is not clear to me. First, if CuSO4 is the source of Cu, the solution should be blue; second, I do not see any reductant for Cu(0)NPs to be formed.

6) Page 6, Fig. 5: the Cu signals in the EDX spectra are not observable. The corresponding regions should be magnified in order to make the signals visible.

7) Page 7, Figures 6 and 7: what is the Cu concentration in the TEM images? Is the particle size similar irrespective of the Cu concentration? I suggest to provide particle distribution graphics for the different samples (as in Fig. 8).

8) Page 10, Figure 9: the text below the bars is too small and cannot be read.

9) The antibiotic activity of the colloidal solution should be also tested for comparison purposes.

Reviewer 3 Report

The source of supply for the bacteria strains is not mentioned in the manuscript.

The type of electro spinner should be specified (only the manufacturer is mentioned).

Scale bars should be added to micrographs (Fig 3, 4).

Larger images with a better quality/resolution should be provided (Fig 4, 9).

The images in Fig 4 represent elemental compositions? This requires further explanation.

Fig 5 should be improved: The scaling does not allow to recognize differences in smaller peaks.

Sample preparation for Fig 6 and 7 should be specified (TEM and EDX of colloidal solutions?).

Error bars should be added to the graphs. Alternatively, experimental uncertainties should be discussed in the text (Fig 8, 9).

For values that were determined experimentally, only significant digits should be given in the manuscript (Table 5 col 4).

Round 2

Reviewer 1 Report

The work adds to the existing repertoire in antibacterial activity of electrospun polyacrylonitrile metal nanoparticle nanofibers.

Reviewer 2 Report

Most of the suggestions by the reviewers remain unattended. Everybody understands the limitations because of the pandemic restrictions, but we must be rigorous during the refereeing process and cannot accept manuscripts that not include enough experimental support to prove the hypotheses. The authors use the pandemic restrictions as an excuse to avoid the suggested experimental work, something that we cannot accept. Therefore, acceptance of the manuscript is conditioned to the authors providing the requested information below:

- Concerning my previous comment “Is the PAN-CuNPs nanofiber just a solid heterogeneous mixture of PAN and CuSO4?”, the authors answer that “CuSO4 was reduced by the DMF after PAN and CuSO4 and were mixed in DMF solution for 24 hours”, and in the manuscript one can read that “color change from colorless to dark green can be observed, signifying the formation, oxidation, and agglomeration of CuNPs in solution”. In the answer to my question regarding the oxidation state of Cu, the authors say “The oxidation state of Cu is Cu(II) because all CuNPs found in PAN-CuNP nanofibers and PAN/DMF/CuNP colloidal solution have formed as a result of the reduction of CuSO4 via DMF solution”. This issue remains unclear and must be properly explained and proven. If there is a reduction of CuSO4, this cannot be the reason for the state of Cu being II; maybe first there is a reduction to Cu(0) and then oxidation to Cu(I) and/or Cu(II) in the presence of air.

As regards the references provided by the authors to support their arguments, Ref. 65 is not an appropriate one; a reference where DMF reduces Cu(II) to Cu(0) must be provided. Ref. 73 involves NaBH4 as reductant, not DMF. In ref. 74, CuNPs are prepared by a biological method. Ref. 75 does not deal with CuNPs but with AuNps and AgNPs.

- As the authors did in ref. 48, the copper leaching must be measured. The oxidation state must be proven someway, XRD or XPS; the color changes indicated by the authors could be also shown in the manuscript with the corresponding pictures.

Reviewer 3 Report

My reviewer concerns were largely considered. The revised version of the manuscript is recommended for publication.
